# Retrospective Analysis of Artifacts in Cone Beam Computed Tomography Images Used to Diagnose Chronic Rhinosinusitis

**DOI:** 10.3390/diagnostics13091511

**Published:** 2023-04-22

**Authors:** Niina Kuusisto, Jussi Hirvonen, Auli Suominen, Stina Syrjänen, Sisko Huumonen, Pekka Vallittu, Ilpo Kinnunen

**Affiliations:** 1Department of Oral Pathology and Radiology, Institute of Dentistry, University of Turku, 20520 Turku, Finland; 2Department of Radiology, Päijät-Häme Central Hospital, 15850 Lahti, Finland; 3Department of Radiology, University of Turku and Turku University Hospital, 20520 Turku, Finland; 4Medical Imaging Center, Department of Radiology, Tampere University and Tampere University Hospital, 33521 Tampere, Finland; 5Department of Community Dentistry, Institute of Dentistry, University of Turku, 20520 Turku, Finland; 6Institute of Dentistry, University of Eastern Finland, 70210 Kuopio, Finland; 7Diagnostic Imaging Center, Kuopio University Hospital, 70029 Kuopio, Finland; 8Research Unit of Oral Health Sciences, University of Oulu, 90570 Oulu, Finland; 9Department of Biomaterials Science and Turku Clinical Biomaterials Centre—TCBC, Institute of Dentistry, University of Turku, 20520 Turku, Finland; 10Welfare Division, City of Turku, 20520 Turku, Finland; 11Department of Otorhinolaryngology-Head and Neck Surgery, Turku University and Turku University Hospital, 20520 Turku, Finland

**Keywords:** cone beam CT, artifact, rhinosinusitis, dental restorations, apical periodontitis, maxilla

## Abstract

Background: Cone beam computed tomography (CBCT) is frequently used to corroborate the signs and symptoms of chronic rhinosinusitis (CRS). However, artifacts induced by dental restorations might complicate the diagnosis of CRS. Here, we assessed the frequency and location of artifacts in CBCT images taken to confirm the CRS. Methods: All CBCT images of the patients referred to the Emergency Radiology unit, Turku University Hospital, with an indication of CRS in 2017 were re-examined. The prevalence of the artifacts was analyzed in three cross-sectional views and three horizontal levels delimited by anatomical landmarks. Results: In total, 214 CBCT images of patients with CRS were evaluated. The diagnosis of apical periodontitis (AP) was impaired by artifacts present in 150/214 images (70%). The diagnosis of CRS was impaired in 5 of the 214 images (2.3%). The main origins of the artifacts were large dental fillings or crowns, and endodontic fillings were present in 95% (203/214) and 52% (111/214) of the images, respectively. Conclusions: AP as an etiology of CRS is possible to miss because of artifacts originating from dental and endodontic fillings in the CBCT images of the paranasal sinuses.

## 1. Introduction

Chronic rhinosinusitis (CRS) is a common disease, being of dental origin in up to 40% of patients [1,2]. Although CRS of dental origin distinctly differs from CRS, it is often misdiagnosed as CRS because of similar symptoms. CRS of dental origin, also called odontogenic sinusitis, does not have a standard code or a clear criterion, which makes the diagnosis even more complex. Initially, CRS of dental origin originates from iatrogenic damage of the mucoperiosteum of the maxillary sinus [3]. Apical periodontitis (AP) is the most common cause of CRS of dental origin, and is an important radiologic sign that may indicate CRS of dental origin [4]. An additional radiological sign is a unilateral mucosal thickening on the maxillary sinus floor. Various dimensions for mucosal thickening have been defined, mostly 2 mm or more [5,6,7]. Although mucosal thickening and AP of the maxillary teeth are correlated, mucosal thickening can be present without symptoms in 29–53% of cases [7,8,9]. Consequently, radiology and clinical examination are both essential in recognizing CRS of dental origin.

Two-dimensional imaging alone is usually insufficient to diagnose the paranasal sinuses and the posterior maxillary teeth. CBCT or CT are recommended imaging methods for the paranasal sinuses, especially when assessing the severity of the infection or anatomical variants, and are required before surgery [4]. Although the modern CBCT equipment has a sufficient image quality, artifacts can cause serious problems. CBCT image artifacts are already well described as a phenomenon, as well as the challenges of image interpretation [10,11,12,13,14,15,16]. Beam hardening (BH) is the most common CBCT image artifact composed of a cupping artifact, hypodense areas, and streaks found when imaging dense materials [17]. Although many different correction algorithms, such as metal artifact reduction algorithms (MAR), are available, no optimal solution for artifact problems in the dento-maxillofacial area has been established [18].

CRS is a multifactorial condition in which the microbiota plays a pathogenic role. The bacterial microbiomes in CRS of dental or non-dental origin are different [19]. Thus, knowing the etiology of CRS is important for successful management. CRS of dental origin usually requires dental surgery or a combination of medical treatment and dental or endoscopic sinus surgery [19,20]. How often the diagnosis of CRS of dental origin remains uncertain due to artifacts is currently unknown. Thus, the present retrospective study aimed to investigate the prevalence of artifacts due to dental restorations, the frequency of artifacts that complicate the diagnosis of CRS, and how often AP can be identified as the possible origin of CRS in the CBCT images of paranasal sinuses.

## 2. Materials and Methods

The search criteria for the CBCT images of paranasal sinuses were patients aged ≥50 years with CRS as the indication for the CBCT imaging. All images taken in the year 2017 (1 January–31 December 2017) at the Emergency Radiology unit in the Turku University Hospital were included. Imaging was performed with Planmed Verity CBCT (Planmed, Helsinki, Finland), with the following settings: 16 kV, 6 mA, FOV of 16 × 13 cm^2^, slice thickness of 0.2 mm, an isotropic voxel size of 0.2 × 0.2 × 0.2 mm^3^, and an acquisition time of 8 s. The permission for this retrospective study was obtained from the Hospital District of Southwest Finland, permission number T06/051/17. The requirement of written patient consent was waived due to the retrospective nature of the study. Cases were identified from the picture archiving and communication systems (PACS) and the radiological information system (RIS) using standard codes of the CBCT of the paranasal sinuses (DM1AI), and data were cross-referenced with patient medical files.

All the CBCT images were re-reviewed by two radiologists: one specialized in oral and maxillofacial radiology and the other in head and neck radiology. The referrals and medical reports were re-checked to ascertain the diagnosis or the suspicion of CRS, and the etiology of the infection. The magnitude of the artifact was analyzed visually and the following question was considered: does it disturb the recognition of the anatomy and can a diagnosis of infection be made? (yes/no). The artifacts were analyzed in three cross-sectional views, axial, sagittal, and coronal, and in three horizontal levels as shown in Figure 1. The horizontal levels were delimited by anatomical landmarks, and in each CBCT image, the artifacts were recorded at all individual levels. An artifact was recorded as complicating the diagnosis, even when the diagnosis was impaired only in one slice of the one cross-sectional view of the CBCT image.

Analyzing the anatomy in the dentoalveolar area of the maxilla, at level 1 (Figure 1), required the recognition of the periodontal ligament space surrounding the premolar and molar roots, and endodontic fillings including the root canal posts were necessitated. Implants and dental crowns were recognized by their structure and shape in the CBCT images. A dental filling on the mesial, occlusal, and buccal sides of the tooth was defined as a large dental filling. Different materials such as amalgam, composite, or ceramic could not be differentiated in the CBCT images. AP of the maxillary premolar or molar was diagnosed when a distinctly widened periodontal ligament or a radiolucent lesion was identified in the periapical area of the root. To analyze the anatomy of the maxillary sinus at level 2 (Figure 1), the cortex of the sinus walls, particularly the maxillary sinus floor, had to be visible. A mucosal thickening of 2 mm or more was needed to make the diagnosis of sinusitis. The same criteria were required for the other paranasal sinuses at level 3 (Figure 1).

Interobserver agreement was calculated using the ICC test (interobserver class correlation coefficient). The SPSS 29 (IBM, Armonk, NY, USA) software package was used.

## 3. Results

In total, 214 patients with CBCT images taken during 2017 were identified with a diagnosis of CRS. Of these patients, 139 were women (65%) and 75 were men (35%), with a median age of 62 years and with a range from 50 years to 89 years. Seven reports of the CBCT images were missing. Based on the reports, CRS of dental origin was suspected in 24/214 (11%) of the CBCT images due to AP diagnosed in the upper jaw (Figure 2). Artifacts were recorded in the reports in only four images, and an additional X-ray imaging examination was recommended twice due to the artifacts.

We re-examined all 214 CBCT images to analyze the location and prevalence of all artifacts, even the minor ones, in more detail. All artifacts were recorded in three cross-sectional views, axial, sagittal, and coronal, and in three horizontal levels, as given in Figure 1. As expected, artifacts of any degree were most common at level 1, present in all 214 images. Level 2 artifacts were less frequent, present in 60/214 (28%) of the CBTC images. At level 3, covering the sphenoid, ethmoid, and frontal sinuses, artifacts were recognized in only one image (0.5%) due to a metal plate in the maxilla (Figure 3). Artifacts were equally seen in all coronal, axial, and sagittal views. We also analyzed the interobserver reliability to detect artifacts. In total, 50 CBCT images were selected randomly to calculate the overall agreement of the interobserver reliability. The overall agreement to identify artifacts at level 1 was almost perfect, 0.960, 95% CI 0.90–1.00 (48 out of 50); at level 2, substantial, 0.560, 95% CI 0.42–0.69 (28 out of 50); and at level 3, also almost perfect 0.940, 95% CI 0.87–1.00 (47 out of 50).

The obstacles of the artifacts are given in Figure 4. The interpretation of the anatomy of the maxillary teeth in detail was hampered in most of the CBCT images, i.e., 208/214 (97%). The detailed exploration of AP was complicated because of artifacts in 150/214 (70%) of the images. Artifacts complicated the recognition of the anatomy of the maxillary sinus in 16/214 (7.5%) of the images. The diagnosis of sinusitis was precluded due to an artifact in 5/214 (2.3%) of the images. Artifacts did not complicate the detailed evaluation of the anatomy of paranasal sinuses nor the diagnosis of paranasal sinusitis at level 3 (0%).

The origin of the artifacts in the CBCT images is given in Figure 5. The main origins of the artifacts were large dental fillings or crowns in the upper jaw, being present in 203/214 (95%) of the CBCT images (Figure 6). Endodontic fillings of the maxillary teeth were the reason for artifacts in 111/214 (52%) of the images (Figure 7), followed by fixed dental prostheses and dental implants (Figure 8) causing the artifacts in 28/214 (13%) and 5/214 (2.3%) of the CBCT images, respectively. Other implants in the maxillofacial area were found only in three CBCT images, resulting in artifacts in levels 2 and 3 (Figure 3).

## 4. Discussion

The diagnosis of apical periodontitis (AP) in the CBCT image of paranasal sinuses can be complicated due to artifacts originating from dental and endodontic fillings. In our present study, we could confirm that the causes of the artifacts in the CBCT images were dental fillings or crowns followed by endodontic fillings. Hence, there is a risk that the diagnosis of CRS originating from AP can be missed, or that AP diagnosis might be delayed. As shown here, the diagnosis of sinusitis of the paranasal sinuses is usually not hindered due to artifacts in CBCT images. However, artifacts complicated the evaluation of apical periodontitis in 70% of the images, disqualifying the diagnosis of CRS of dental origin.

CRS of dental origin is most prevalent in the age group of 40–60 years and is more often found in women [3,21]. This is why we selected only those patients aged ≥50 years, referring to CBCT imaging during the year 2017, with an indication of possible CRS. In our study, women with CRS were more prevalent than men, being in line with previous studies [3,21]. Here, we found that CRS was suspected to be associated with AP in 11% of the 214 CBCT images. The artifacts were rarely mentioned in the CBCT reports, although they hampered the diagnosis of AP in 70% of the images when re-evaluated. Based on the CBCT reports, additional X-ray images of the maxillary teeth, such as the intraoral ones, were not recommended despite difficulties in assessing the periapical region due to artifacts. Describing the artifacts and their detrimental effects regarding the diagnosis of the periapical area of the tooth is of importance for clinicians to evaluate the need for additional X-ray images, especially in patients with substantially restored dentition. It has to be mentioned, however, that additional X-ray images examined by other organizations may have been missed in this study.

We showed here that the artifacts were equally seen in all the cross-sectional views. Schulze and coworkers reported that axial slices showed artifacts more intensely because of the beam direction [22]. Interpreting the artifacts in the CBCT images is, after all, subjective and is dependent on the experience specifically in this anatomical region, including the morphology of different teeth. In our study, the overall agreement between two observers was the lowest at level 2, but still remained substantial. This might be due to the challenging area of the maxillary sinus floor which varies individually. Additionally, the oral and maxillofacial radiologists who graduated first with a Bachelor of Dental Sciences (BDS) might have more interest in assessing the periapical region of premolar and molar teeth, with respect to the orientation of individual roots.

In re-evaluating the 214 CBCT images, we found that artifacts are common in elderly patients, as most of them have had restorative dentition treatments with a wide range of materials used. As an example, the CBCT image given in Figure 6 illustrates the detrimental effect of dental restorations. The magnitude of artifacts in CBCT images caused by materials used in dentistry depends on the atomic weight of the material as well as the size and thickness of the object [22]. According to the literature, titanium implants are the source of detrimental artifacts especially in the CBCT images [15,23,24], as are also the Zirconia implants [25,26,27]. Fiber-reinforced composites (FRC), which are clinically used in cranioplasty surgery, seem to be without detrimental effects in the CBCT images [28]. We have shown earlier that composite materials which consist of at least 20% radio-opacifying BaAlSiO2 fillers can cause artifacts in the CBCT images [29]. In the present study, there were only a few patients with dental (5/214) and maxillofacial implants (3/214), but as shown here, multiple dental implants cause deleterious artifacts (Figure 8).

AP is an inflammatory disorder of the tooth root caused by bacterial invasion of the pulp, such as in untreated dental caries. At first, AP appears as a widened periodontal ligament in the apex of the root, and possibly in only one of the roots of a molar. Therefore, evaluating AP in the CBCT image of paranasal sinuses requires a careful examination of each root considering the root anatomy. AP is the most common cause of CRS of dental origin due to the close relationship between the maxillary molar roots and the maxillary sinus floor (Figure 2). This study focused only on AP as a cause of CRS. However, other dental diseases can cause CRS, such as periodontitis, vertical bone loss, and endodontic-periodontic pathology, as well as iatrogenic causes, such as oroantral fistula, foreign bodies, misplaced roots, dental and root canal fillings, and sinus lift procedures [4]. These causes indicate that CRS of dental origin is not only a wide problem among those who cannot afford dental treatments, but also among those who can afford extensive treatments such as dental implants [20,30].

Here, we also showed that endodontic fillings such as gutta-percha and root canal posts caused artifacts in 52% of the CBCT images (Figure 7). However, their role in causing detrimental artifacts and in diagnosing AP remains partly unknown. When assessing root-filled teeth, it is difficult to judge whether the artifact originates more from the crown material, the root filling, or the root canal post material. Regardless of the origin, root-filled teeth should be checked extra carefully, as AP is more common in root-filled teeth, and teeth restored with crowns [31,32,33,34]. Furthermore, periapical lesions are also more common in molar teeth and the mesiobuccal roots of maxillary first molars [31]. Maillet and coworkers reported that the maxillary first molar was the tooth associated most with a change in the maxillary sinus floor (55%), followed by the maxillary second molar (34%), the second premolar (8%), and the first premolar (3%). This order is most likely a consequence of the close location of the maxillary molar roots to the maxillary sinus floor and the earlier eruption of the maxillary first molar, causing a greater risk of carious lesions [35].

Notably, artifacts can vary between CBCT vendors, especially when imaging root-filled teeth, as shown by Vasconcelos and coworkers [36]. The newest CBCT machines have metal artifact reduction (MAR) methods, which are algorithms applied in tomographic image reconstruction to minimize artifacts. However, they can efficiently improve image quality only when more pronounced and strong artifacts are present [37,38]. In our study, MAR methods were unavailable in the CBCT equipment. In addition to artifacts caused by material, there are multiple other artifacts hampering the evaluation of CBCT images, such as motion artifacts, extinction artifacts, the partial volume effect, aliasing artifacts, and ring artifacts [22]. Many of these can be corrected by calibrations depending on the CBCT equipment and software. To enhance the image quality and reduce artifacts, the field of view can be adjusted to exclude the crowns of the maxillary teeth or implants in the CBCT image [39]. According to Pauwels and coworkers, diagnosing the area between metal objects should be avoided, as well as regions close to the metal in CBCT images. Additionally, voxel value measurements should be avoided [24].

Due to artifacts, many authors still recommend intraoral images especially when dental implants are present, as seen in Figure 8 [40]. Therefore, a careful clinical judgment of the magnitude and materials used in dental restoration should be performed before referring patients for CBCT imaging.

Our study clearly showed how different materials used in dental fillings can hinder or delay the diagnosis of AP, which is one of the main causes of CRS. Thus, our results are in line with earlier studies, as reviewed by Terrabuio and collaborators in 2021 [41]. Demirturk Kocasarac et al. showed in their phantom study that images with smaller voxel sizes showed a greater number of artifacts, with the 0.2 mm voxels resulting in the highest number of artifacts [42]. We used, in our study, only 0.2 mm voxels, which are detailed and accurate. However, the risk for increased artifacts should be considered when imaging with small voxel sizes. Thus, the results of both Demirturk Kocasarac et al. and our study might indicate the use of higher voxel sizes when imaging patients with extensive dental treatment. However, there are several aspects to take into consideration before definitive conclusions are made, such as the differences in the CBCT machines used in imaging. Furthermore, the images taken of patients differ from phantom images, e.g., due to the absence of soft tissues.

## 5. Conclusions

In conclusion, the diagnosis of AP can be complicated in 70% of the CBCT images of paranasal sinuses due to artifacts originating from dental and endodontic fillings. According to these results, it is possible to miss AP as an etiology of CRS because of artifacts. In agreement with the previous studies, when evaluating patients with CRS, dental etiology should be suspected, especially in those cases with previous dental treatments.

## Figures and Tables

**Figure 1 diagnostics-13-01511-f001:**
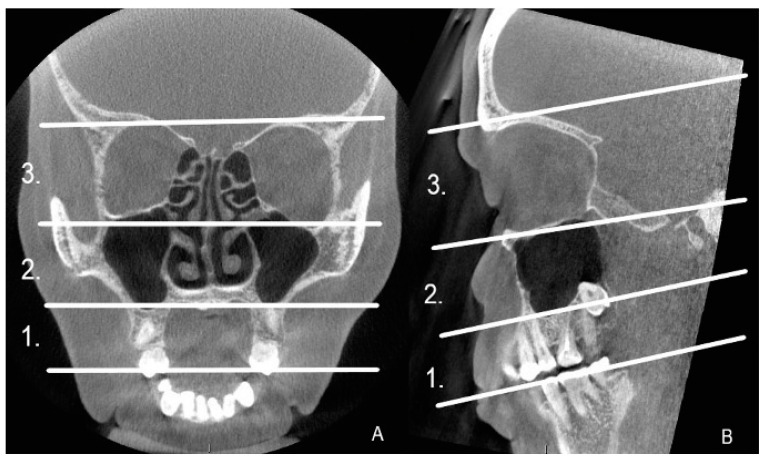
Coronal (**A**) and sagittal (**B**) view of the CBCT image divided in levels 1–3. 1. The area of the premolar-molar teeth and the alveolar ridge of the maxilla. 2. The area from the maxillary sinus floor to the orbital floor. 3. The area from the orbital floor to the orbital roof.

**Figure 2 diagnostics-13-01511-f002:**
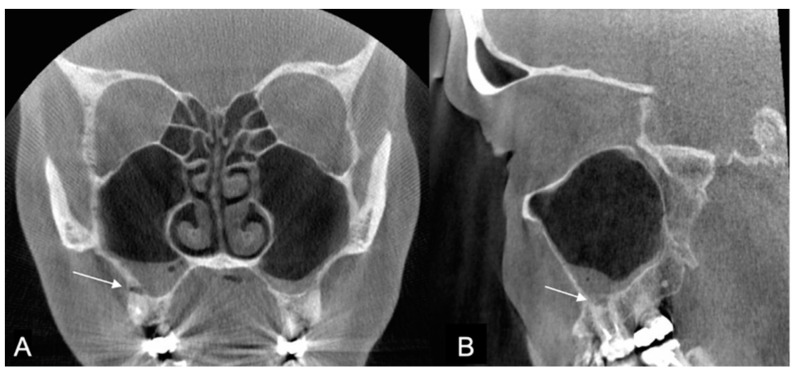
The coronal (**A**) and sagittal (**B**) slices of the CBCT image present sinusitis and distinct apical periodontitis of the root-filled right maxillary molar D16 buccal roots (arrow).

**Figure 3 diagnostics-13-01511-f003:**
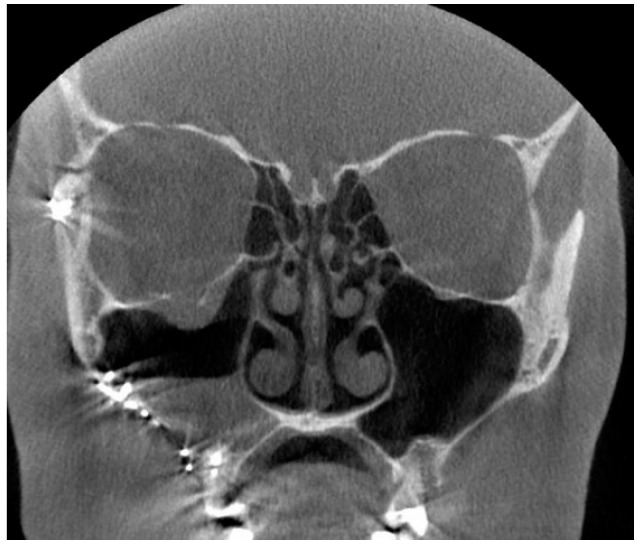
The coronal slice of the CBCT image shows artifacts induced by a metal plate on the right side of the maxilla.

**Figure 4 diagnostics-13-01511-f004:**
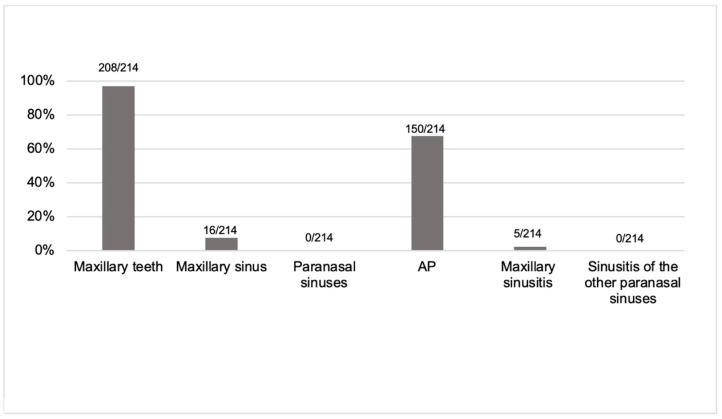
The prevalence of the CBCT images, where artifacts hampered either the evaluation of a given anatomic region or the diagnosis of apical periodontitis (AP) or sinusitis.

**Figure 5 diagnostics-13-01511-f005:**
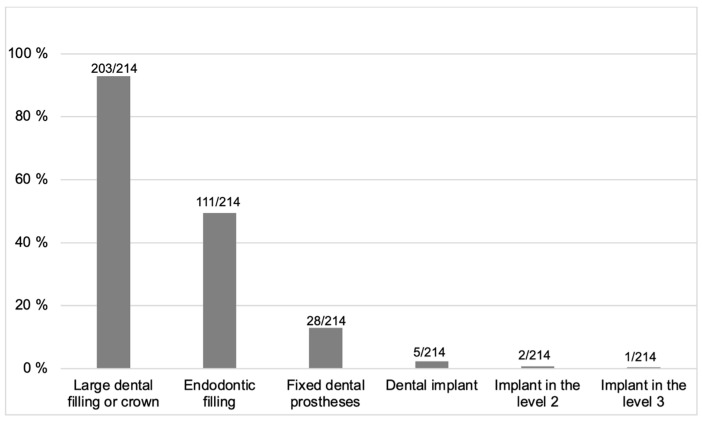
The origins of the artifacts.

**Figure 6 diagnostics-13-01511-f006:**
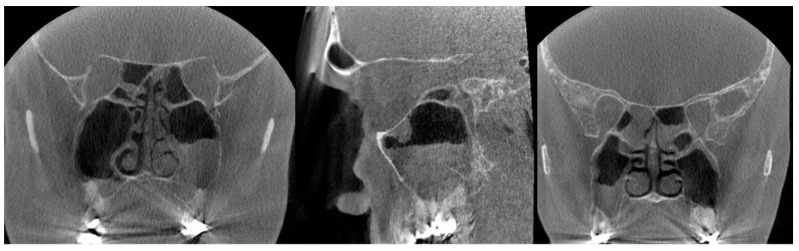
(**Left**): The coronal and sagittal CBCT slices present the signs of sinusitis and artifacts induced by dental restorations in the maxilla. Possible apical periodontitis cannot be distinguished accurately because of the artifacts. (**Right**): The coronal slice of CBCT image presenting mucosal thickening in the paranasal sinuses and artifacts induced by dental restorations hampering the evaluation of the periapical region of the teeth in detail.

**Figure 7 diagnostics-13-01511-f007:**
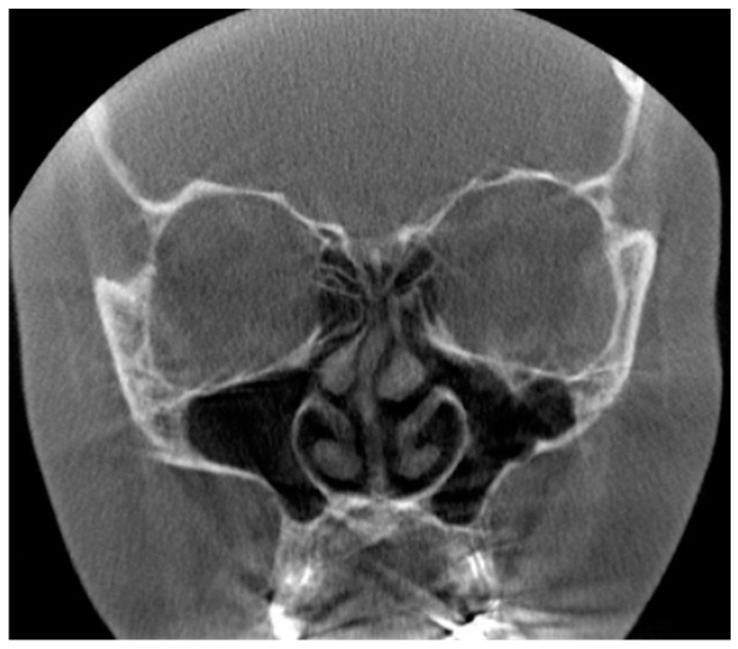
The coronal slice of the CBCT image presenting root-filled teeth in the maxilla and some artifacts originating from the fillings.

**Figure 8 diagnostics-13-01511-f008:**
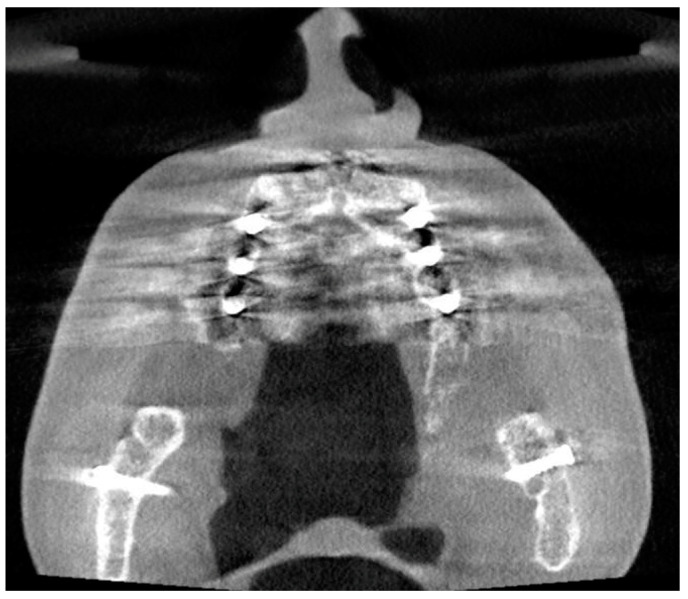
The axial slice of the CBCT image presents multiple implants causing a substantial amount of artifacts.

## Data Availability

Data is contained within the article.

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
