# Peer review of "Retrospective Analysis of Artifacts in Cone Beam Computed Tomography Images Used to Diagnose Chronic Rhinosinusitis"

_diagnostics, 2023, doi:10.3390/diagnostics13091511_

Round 1

Reviewer 1 Report

The manuscript addresses a modern and multidisciplinary research topic with an impact on chronic rhinosinusitis (CRS) diagnostic. The authors investigate the prevalence of artifacts

on CBCT (cone beam computed tomography) images due to dental restorations. Also, they investigated the frequency of artifacts that complicate the diagnosis of CRS, and how often dental apical periodontitis (AP) can be identified as the possible origin of CRS The originality of the research is represented by the fact the authors proposed to observe how often the diagnosis of CRS of dental origin remains uncertain due to artifacts, this issue being still uncertain. The research conclusion showed that the diagnosis of AP can be complicated in 70 % of the CBCT images of paranasal sinuses due to artifacts originating from dental and endodontic fillings.  The researchers recommended careful clinical judgment to CBCT imaging to reach the correct diagnosis of CRS of dental origin without delay.

Author Response

Thank you for your improvement suggestions. We have revised the manuscript according to your comments.

I think it would be better if the authors mentioned at the beginning of the”Material and method” chapter that the study is retrospective.

  • Retrospective study is already mentioned in the lines 69, 80 and 81 as well as in the title.

I think it would be better if the authors more clearly specified the statistically analyzed data in the “Material and method” chapter.

  • We revised this in the chapter materials & methods

I think it would be better if figures 6, 7 and 8 were included in the “Results” chapter to exemplify a certain type of result.

  • Figures 6-8 are now mentioned in the results and the figures 7 and 8 were changed their order.

I believe that the results of the statistical analysis performed with SPSS should be more clearly and in detail presented.

  • We revised this in the chapter materials & methods

Reviewer 2 Report

The manuscript is well written. The conclusions in the abstract section and the manuscript do not match and should be improved. The manuscript may be published after incorporating these suggestions.

Author Response

Thank you for your improvement suggestions. We have revised the manuscript according to your comments.

The manuscript is well written. The conclusions in the abstract section and the manuscript do not match and should be improved. The manuscript may be published after incorporating these suggestions.  
  • We revised the abstract and the conclusions chapter to be convergent.

Reviewer 3 Report

This article clearly explains how different materials used in dental fillings can hinder or delay the diagnosis of AP, one of the main causes of CRS. Data presented here are very similar to data found in literature (Terrabuio BR, Carvalho CG, Peralta-Mamani M, Santos PSDS, Rubira-Bullen IRF, Rubira CMF. Cone-beam computed tomography artifacts in the presence of dental implants and associated factors: an integrative review. Imaging Sci Dent. 2021 Jun;51(2):93-106. doi: 10.5624/isd.20200320. Epub 2021 Mar 11. PMID: 34235055; PMCID: PMC8219451)

According to Demirturk Kocasarac et al (Demirturk Kocasarac H, Ustaoglu G, Bayrak S, Katkar R, Geha H, Deahl ST, 2nd, et al. Evaluation of artifacts generated by titanium, zirconium, and titanium-zirconium alloy dental implants on MRI, CT, and CBCT images: a phantom study. Oral Surg Oral Med Oral Pathol Oral Radiol. 2019;127:535–544), images with smaller voxel sizes showed a greater amount of artifacts: the 0.2-mm voxels have the most number of artifacts. This study used only 0.2 mm voxels which are more detailed and accurate, but they are at risk of showing artifacts. I think we should ask ourselves if with patients with dental treatment it would be worth using higher sizes

It also would be nice adding more images of CBCT showing artifacts caused by different endodontic materials

Author Response

Thank you for your improvement suggestions. We have revised the manuscript according to your comments.

We added the references in the discussion chapter and reflect the voxel size and artifacts.

Unfortunately, images of different endodontic fillings are not available.